# Chemerin Treatment Inhibits the Growth and Bone Invasion of Breast Cancer Cells

**DOI:** 10.3390/ijms21082871

**Published:** 2020-04-20

**Authors:** Hyungkeun Kim, Joo-Hee Lee, Sun Kyoung Lee, Na-Young Song, Seung Hwa Son, Ki Rim Kim, Won-Yoon Chung

**Affiliations:** 1Department of Applied Life Science, The Graduate School, Yonsei University, Seoul 03722, Korea; khg165@gmail.com (H.K.); lpluto@yuhs.ac (S.K.L.); nysong608@yuhs.ac (N.-Y.S.); 2Department of Oral Biology and BK21 PLUS Project, Yonsei University College of Dentistry, Seoul 03722, Korea; jhlee1201@yuhs.ac (J.-H.L.); shson0729@yuhs.ac (S.H.S.); 3Oral Cancer Research Institute, Yonsei University College of Dentistry, Seoul 03722, Korea; 4Department of Dental Hygiene, College of Science and Engineering, Kyungpook National University, Sangju 37224, Korea; rim0804@knu.ac.kr

**Keywords:** chemerin, breast cancer cell invasion, EMT, RANKL/OPG, bone resorption

## Abstract

Chemerin is secreted as prochemerin from various cell types and then cleaved into the bioactive isoform by specific proteases. In various cancer types, chemerin exhibits pro- or antitumor effects. In the present study, chemerin treatment significantly inhibited the viability and invasion of breast cancer cells in the absence or presence of transforming growth factor (TGF)-β and insulin-like growth factor (IGF)-1. The expression levels of E-cadherin and vimentin were reduced in chemerin-treated breast cancer cells. However, chemerin treatment recovered the reduced E-cadherin expression level in breast cancer cells treated with TGF-β or IGF-1. Chemerin treatment inhibited nuclear β-catenin levels in breast cancer cells stimulated with or without TGF-β or IGF-1. In addition, chemerin treatment blocked the increase in the receptor activator of nuclear factor kappa-Β ligand (RANKL)/osteoprotegerin (OPG) ratio in osteoblastic cells exposed to metastatic breast cancer cell-derived conditioned medium. Chemerin treatment inhibited RANKL-induced osteoclast formation and bone resorption by reducing the secretion of matrix metalloproteinase (MMP)-2, MMP-9, and cathepsin K. Intraperitoneal administration of chemerin inhibited tumor growth in MCF-7 breast cancer cell-injected mice and reduced the development of osteolytic lesions resulting from intratibial inoculation of MDA-MB-231 cells. Taken together, chemerin inhibits the growth and invasion of breast cancer cells and prevents bone loss resulting from breast cancer cells by inhibiting finally osteoclast formation and activity.

## 1. Introduction

Chemerin is a small 16-kDa protein with chemoattractive activity toward chemerin/chemokine-like receptor (CMKLR)1-expressing immune cells, including natural killer, macrophage, and dendritic cells [1]. Chemerin is secreted as prochemerin from various cell types and then cleaved into bioactive isoforms by specific proteases [2]. Chemerin is also considered to function as an antimicrobial agent in the epidermis, an angiogenic factor in endothelial cells, and an adipokine that regulates adipogenesis and energy metabolism in adipocytes [3,4,5].

Recent studies have brought attention to the correlation between chemerin and cancer and the potential use of chemerin as a diagnostic and prognostic marker in most cancer types. Interestingly, chemerin has shown different effects on different cancer types. Chemerin levels were increased in the sera of patients with gastric cancer [6], adrenocortical carcinoma [7], early stage oral squamous cell carcinoma [8], non-small cell lung cancer (NSCLC) [9,10], colorectal cancer [11], and pancreatic cancer [12] compared with the respective healthy controls. The elevated serum level of chemerin was positively associated with advanced clinical stages and poorer overall and disease-free survivals in patients with gastric cancer or NSCLC [6,9] but with improved overall survival in patients with adrenocortical carcinoma [7]. The expression level of the chemerin gene or protein was downregulated in the tissues of patients with hepatocellular carcinoma (HCC) [13], adrenocortical carcinoma [14], melanoma, lung cancer, prostate cancer, and colon cancer [15], and increased expression of chemerin correlated with improved clinical outcomes in patients with HCC, adrenocortical carcinoma, or melanoma.

The pro- or antitumor effect of chemerin and its underlying mechanisms may also be context-dependent. Treatment with human recombinant chemerin promoted the invasiveness of gastric cancer cells by upregulating vascular endothelial growth factor, matrix metalloproteinase (MMP)-7 and interleukin-6 via phosphorylation of p38 and extracellular signal-regulated kinase1/2 MAPK [16] and by reducing the secreted levels of tissue inhibitors of metalloproteinase (TIMP)-1 and TIMP2 via CMKLR1 and G protein-coupled receptor 1 [17]. Chemerin derived from cancer-associated myofibroblasts stimulated cell invasion and increased the secreted levels of MMP-1, MMP-2, and MMP-3 in CMKLR1-expressing esophageal squamous cancer cells [18]. In several cancer types, the antitumor effects of chemerin have been linked to increased recruitment of innate immune defense components, natural killer and T cells [15,19] or suppression of the inflammatory tumor microenvironment and myeloid-derived suppressive cell accumulation [20]. In addition, overexpression of chemerin inhibited the in vitro migration and invasion and in vivo metastasis of HCC cells by blocking the phosphorylation of Akt via the CMKLR1-PTEN interaction [21]. In adrenocortical carcinoma, chemerin overexpression decreased cell proliferation and invasion in vitro and tumor growth in vivo by promoting β-catenin phosphorylation and degradation and inhibiting the phosphorylation of p38 MAPK [14].

Although breast cancer is the most common malignancy in women worldwide, and an increased risk of breast cancer is closely associated with postmenopausal obesity, the role of chemerin in breast cancer remains unclear. Chemerin gene expression was significantly downregulated in breast cancer tissues compared with that in normal tissues according to analysis of datasets from the Gene Expression Omnibus (GEO) database [15]. The forced expression of chemerin in breast carcinoma led to the recruitment of antitumor immune cells into the tumor microenvironment [19]. On the other hand, a recent study reported that chemerin was highly expressed in breast cancer tissues compared with adjacent normal tissues and that chemerin expression was significantly correlated with weight, body mass index, tumor growth, metastasis, and poor overall survival [22]. In the present study, we attempted to determine the effect of exogenously treated chemerin on breast cancer cell growth and invasion and related bone resorption.

## 2. Results

### 2.1. Chemerin Inhibited the Viability and Invasion of Breast Cancer Cells

To determine the effect of chemerin on breast cancer cells, we first investigated its effect on the viability, migration, and invasion of MDA-MB-231 and MCF-7 cells. Treatment with 80 nM chemerin for 24 h and 72 h inhibited the viability of MDA-MB-231 cells by 11% and 17%, respectively. The migration of MDA-MB-231 cells was increased by treatment with 80 nM chemerin for 24 h, but cell invasion was inhibited by chemerin treatment in a dose-dependent manner; in particular, it was reduced by 53% in cells treated with 80 nM chemerin for 24 h (Figure 1A). In MCF-7 cells treated with 80 nM chemerin for 24 h and 72 h, cell viability was inhibited by 11% and 24%, respectively. Cell migration was not changed, but cell invasion was inhibited by 74% in MCF-7 cells treated with 80 nM chemerin for 24 h (Figure 1B). The in vitro inhibitory effect of chemerin on breast cancer cell invasion was further confirmed using a chick chorioallantoic membrane (CAM) assay, an in ovo model which was used to study cell invasion [23]. The fluorescence from MDA-MB-231 and MCF-7 cells that had invaded below the CAM surface was reduced by chemerin treatment in a dose-dependent manner. Treatment with 80 nM chemerin inhibited cell invasion by 78% in MDA-MB-231 cells and 39% in MCF-7 cells (Figure 1C).

Epithelial–mesenchymal transition (EMT) and extracellular matrix-degrading proteinases play critical roles in the invasion and metastasis of breast cancer cells [24,25]. To determine whether chemerin treatment influences EMT in breast cancer cells, we investigated the expression level of E-cadherin as an epithelial marker and those of vimentin and β-catenin as mesenchymal markers in MDA-MB-231 and MCF-7 cells exposed to chemerin. Western blot analysis indicated that chemerin treatment reduced E-cadherin and vimentin expression levels in MDA-MB-231 and MCF-7 cells (Figure 1D). Cytosolic levels of β-catenin were increased, and its nuclear levels were decreased by chemerin treatment in both breast cancer cell lines (Figure 1E). We further detected the reduced levels of pro MMP-2 and pro MMP-9 in the conditioned media of chemerin-treated MDA-MB-231 cells and pro MMP-9 in the conditioned media of chemerin-treated MCF-7 cells. Pro MMP-2 was not detected by gelatin zymography with the conditioned media of MCF-7 cells (Figure 1F). These results indicate that chemerin inhibits the invasion and EMT of breast cancer cells. The increased migration of MDA-MB-231 cells may be attributed to the substantial decrease in E-cadherin expression.

### 2.2. Chemerin Suppressed Growth Factor-Induced Cancer Invasion

TGF-β and IGF-1 are known to induce EMT and the invasion of cancer cells. In particular, TGF-β and IGF-1 released from resorbed bone matrix stimulate the growth and invasion of bone metastases in bone microenvironment [26,27]. TGF-β treatment for 72 h showed a tendency to reduce the viability of MDA-MB-231 cells. Treatment with 80 nM chemerin for 72 h reduced the viability of TGF-β-treated MDA-MB-231 cells by 32%. Cell invasion was increased by 1.49-fold by TGF-β treatment for 24 h, but the increase in cell invasion by TGF-β was inhibited by 29% and 63% by treatment with 40 nM and 80 nM chemerin, respectively (Figure 2A). In MCF-7 cells, treatment with TGF-β alone or together with 80 nM chemerin reduced cell viability by 28% and 36%, respectively. Cell invasion was increased by 2.23-fold by TGF-β treatment, but TGF-β-stimulated cell invasion was inhibited by 18% and 22% by treatment with 40 nM and 80 nM chemerin, respectively (Figure 2B). Confocal images of immunostained MDA-MB-231 (Figure 2C) and MCF-7 cells (Figure 2D) indicated that TGF-β treatment downregulated the expression of E-cadherin and stimulated the nuclear translocation of β-catenin and SMAD2/3. Treatment with SB525334 (TGF-β type I receptor inhibitor) or chemerin rescued the expression of E-cadherin detected along the plasma membranes and inhibited the translocation of β-catenin and SMAD2/3 into the nucleus in TGF-β-treated breast cancer cells. In addition, Western blot analysis indicated that chemerin treatment inhibited the phosphorylation of SMAD2/3 in TGF-β-stimulated breast cancer cells (Figure 2E).

Treatment with insulin-like growth factor (IGF)-1 for 72 h increased the viability of MDA-MB-231 cells by 1.24-fold, but 80 nM chemerin treatment reduced the IGF-1-induced increase in viability by 14%. Cell invasion was increased by 1.17-fold in MDA-MB-231 cells by IGF-1 treatment for 24 h, but the effect of IGF-1 on MDA-MB-231 cell invasion was reduced by 16% and 29% by treatment with 40 nM and 80 nM chemerin, respectively (Figure 3A). IGF-1 treatment for 72 h increased the viability of MCF-7 cells by 1.99-fold, but treatment with 80 nM chemerin reduced the IGF-1-induced increase in cell viability by 28%. Cell invasion was increased by 1.48-fold in IGF-1-treated MCF-7 cells after 24 h, but IGF-1-induced cell invasion was inhibited by 24% and 54% by treatment with 40 nM and 80 nM chemerin, respectively (Figure 3B). In MDA-MB-231 (Figure 3C) and MCF-7 cells (Figure 3D), immunofluorescence images showed that IGF-1 treatment reduced E-cadherin expression and stimulated nuclear translocation of β-catenin. Treatment with ZSTK474 (PI3K inhibitor), PD98059 (MEK1/2 inhibitor), or chemerin reversed these effects of IGF-1 by elevating the levels of E-cadherin and β-catenin detected along the membrane. These results indicate that chemerin inhibits invasion and EMT via TGF-β and IGF-1 signaling in both breast cancer cell lines.

### 2.3. Chemerin Decreased the Receptor Activator of Nuclear Factor Kappa-Β Ligand (RANKL)/Osteoprotegerin (OPG) Ratio in Osteoblastic Cells Treated with Conditioned Medium from Metastatic Breast Cancer Cells

Metastatic breast cancer MDA-MB-231 cells induce osteolytic bone metastasis by promoting RANKL production in osteoblasts and the resulting osteoclastogenesis [28]. We investigated RANKL and OPG production in human osteoblastic hFOB1.19 cells treated with conditioned medium from MDA-MB-231 breast cancer cells and/or chemerin. Chemerin treatment for 6 h and 24 h did not affect the viability of osteoblastic cells (Figure 4A). Western blot analysis showed that treatment with the conditioned medium upregulated RANKL expression and downregulated OPG expression in osteoblastic cells. Chemerin treatment inhibited RANKL expression and increased OPG expression in osteoblastic cells treated with the conditioned medium (Figure 4B). Significantly elevated RANKL and decreased OPG levels were observed in the culture medium of osteoblastic cells treated with the conditioned medium of breast cancer cells. Chemerin treatment led to the recovery of the secreted levels of RANKL and OPG to control levels (Figure 4C). These results indicate that chemerin treatment blocks the increase in the RANKL/OPG ratio in osteoblastic cells stimulated with conditioned medium from MDA-MB-231 breast cancer cells, thereby preventing RANKL-induced osteoclastogenesis and breast cancer cell-associated bone loss.

### 2.4. Chemerin Suppressed RANKL-Induced Osteoclast Differentiation and Activity in Bone Marrow-Derived Macrophages (BMMs)

Osteolytic bone metastasis of breast cancer has finally been linked with osteoclast-mediated bone resorption [29]. RANKL treatment stimulated osteoclastogenesis and bone resorption in BMMs as osteoclast precursors. Chemerin treatment did not affect the viability of BMMs (Figure 5A) but significantly inhibited RANKL-induced osteoclast formation by 23% at 80 nM (Figure 5B). In addition, chemerin treatment slightly reduced the formation of resorption pits by mature osteoclasts on inorganic crystalline calcium phosphate-coated plates (Figure 5C) and inhibited the secretion of MMP-2, MMP-9 (Figure 5D), and cathepsin K (Figure 5E), which degrade the organic bone matrix. These results indicate that chemerin suppresses RANKL-induced osteoclast differentiation and the bone-resorbing activity of mature osteoclasts.

### 2.5. Chemerin Suppressed Tumor Growth and Bone Loss in Breast Cancer Cell-Inoculated Mice

The in vivo effect of chemerin on tumor growth and bone invasion was further studied using a tumor xenograft model, in which MCF-7 cells were inoculated into the right flanks of mice, and an intratibial model, in which MDA-MB-231 cells were injected into the medullary cavity of the proximal tibia of mice. In MCF-7 cell-inoculated mice implanted with 17β-estradiol pellets, intraperitoneal administration of chemerin noticeably reduced tumor volume (Figure 6A). In MDA-MB-231 cell-inoculated mice, radiophotographic and μCT 3D images showed that the development of severe osteolytic lesions was prevented by chemerin treatment (Figure 6B). The serum levels of the bone turnover markers calcium, C-terminal cross-linking telopeptide of type I collagen (CTX), and TRAP were increased by MDA-MB-231 inoculation but reduced to control levels by chemerin administration (Figure 6C). These results indicate that chemerin treatment can inhibit the growth and osteolysis of breast cancer cells.

## 3. Discussion

A growing body of epidemiological evidence has demonstrated a relationship between specific circulating adipokines and the risk of obesity-associated cancer, including prostate, colon, and breast cancer [30,31,32]. Most adipokines, including leptin, visfatin, and resistin, have been recognized to activate intracellular signaling networks that contribute to breast malignancy, while adiponectin has been considered to be anticarcinogenic [33,34]. Although conflicting evidence regarding chemerin expression and survival outcomes in breast cancer patients has accumulated, the role of chemerin expression in breast cancer cells has not been deeply studied. In the present study, we observed the potential protective effects of chemerin on breast cancer cell growth and invasion, including bone invasion promoted by interactions between cancer cells, osteoblasts, and osteoclasts. 

Chemerin is secreted as prochemerin, which contains 143 amino acids (18 kDa), from white adipose tissue, liver and lung and is ubiquitously detected in plasma at nanomolar concentrations. Prochemerin is cleaved into chemerin isoforms with varying lengths, receptor affinity, and biological activity through C-terminal cleavage by serine proteases [2,35]. We treated metastatic MDA-MB-231 and less metastatic MCF-7 breast cancer cells with the most active form of chemerin, chem157S. Chemerin treatment more significantly inhibited cell invasion than cell viability in breast cancer cells in the presence or absence of TGF-β and IGF-1, which are growth factors that stimulate cancer cell invasion and are released from the bone matrix by osteolysis. Chemerin treatment seemed to inhibit better TGF-β-induced invasion in MDA-MB-231 cells than in MCF-7 cells. This result may be related with the inhibitory effect of chemerin on TGF-β-induced phosphylation of SMAD2/3 in MDA-MB-231 cells than in MCF-7 cells.

E-cadherin, a cell-surface protein, facilitates epithelial cell-cell adhesion and is connected to the cytoskeleton via β-catenin. Reduced E-cadherin expression during cancer progression results in the release of β-catenin into the cytosol, and the translocation of β-catenin into the nucleus induces the transcription of EMT genes. Increased expression of vimentin, an intermediate filament protein that is associated with a migratory phenotype, is also used as an EMT marker in cancer [36]. In our study, the expression of E-cadherin and vimentin was reduced in breast cancer cells treated with chemerin alone. In our preliminary study, chemerin treatment did not change the expression levels of E-cadherin-regulating transcription factors, Snail, Twist, and Zeb1/2. The decrease in E-cadherin expression may be due to its degradation by any proteases and associated with the increased migration of MDA-MB-231 cells. However, chemerin treatment recovered the reduced E-cadherin expression in breast cancer cells treated with TGF-β or IGF-1. Chemerin effect blocking TGF-β-induced reduction of E-cadherin expression may be more potent than the effect of chemerin itself reducing E-cadherin expression level. In addition, chemerin treatment reduced the nuclear levels of β-catenin in breast cancer cells either stimulated with or not treated with TGF-β or IGF-1. NF-κB-regulated MMP-2 and MMP-9 secretion is also closely associated with EMT [37,38]. The secreted levels of pro MMP-2 and pro MMP-9 were reduced by chemerin treatment. Therefore, chemerin treatment may suppress the invasion of MDA-MB-231 and MCF-7 breast cancer cells by inhibiting EMT in breast cancer cells.

In breast cancer bone metastasis, the release of osteolytic factors from breast cancer cells exacerbates cancer cell-mediated bone loss by increasing the RANKL/OPG ratio in osteoblastic cells and consequently causing osteoclast-mediated bone resorption [39]. Adipocytes, one of the most abundant cell types found in bone marrow, account for approximately 15% of the bone marrow volume in young adults and up to 60% in adults over 65 [40]. Chemerin can be released from adipocytes in bone-tumor microenvironment. In this study, RANKL protein expression and secretion were increased, whereas OPG protein expression and secretion were decreased in hFOB1.19 osteoblastic cells exposed to the conditioned medium of metastatic breast cancer MDA-MB-231 cells, leading to an increased RANKL/OPG ratio. Chemerin treatment reversed the effect of the conditioned medium derived from breast cancer cells on RANKL and OPG. Furthermore, chemerin treatment blocked RANKL-induced osteoclast formation and activation by inhibiting the formation of resorption pits and the secretion of bone matrix-degrading proteases, MMPs and cathepsin K. Overall, chemerin treatment may prevent bone resorption by reducing the RANKL/OPG ratio in osteoblastic cells, suppressing RANKL-induced osteoclastogenesis in osteoclast precursors, and inhibiting osteoclast activity.

The in vitro anti-cancer and anti-bone resorptive effects of chemerin on breast cancer cells were supported by in vivo experiments using murine tumor xenograft and intratibial models. Intraperitoneal administration of chemerin inhibited the growth of MCF-7 breast cancer cells and the development of osteolytic lesions in mice with intratibial inoculation of MDA-MB-231 cells.

In conclusion, chemerin inhibits the growth and invasion of breast cancer cells. In addition, chemerin prevents breast cancer cell-mediated bone loss by reducing the RANKL/OPG ratio in osteoblastic cells and inhibiting osteoclast formation and activity. Chemerin treatment or expression induction may be a promising strategy for inhibiting and treating breast cancer and the resulting bone resorption.

## 4. Materials and Methods

### 4.1. Reagents

Dulbecco’s modified Eagle medium (DMEM), minimum essential medium-alpha (α-MEM), Dulbecco’s modified Eagle medium:nutrient mixture F-12 (DMEM/F-12) without phenol red, Dulbecco’s phosphate buffered saline (PBS), fetal bovine serum (FBS), Hank’s balanced salt solution (HBSS), antibiotic-antimycotic mixture containing 100 U/mL penicillin and 100 U/ mL streptomycin, Opti-MEM, Geneticin (G418), and 0.25% trypsin-EDTA were purchased from Gibco BRL (Grand Island, NY, USA). Recombinant human chemerin (Glu21-Ser157, with an N-terminal Met), mouse soluble receptor activator of nuclear factor kappa beta (RANK) ligand (RANKL), and macrophage colony-stimulating factor (M-CSF) were purchased from R&D Systems (Minneapolis, MN, USA). Anti-human E-cadherin (sc-8426), β-catenin (sc-1496-R), vimentin (sc-6260), lamin B (sc-6216), RANKL (sc-377079), osteoprotegerin (OPG; sc-71747), Smad2/3 (sc-133098), and GAPDH (sc-32233) antibodies were purchased from Santa Cruz Biotechnology (Santa Cruz, CA, USA). Transforming growth factor (TGF)-β and insulin-like growth factor (IGF)-1 were purchased from Peprotech (Rocky Hill, NJ, USA). Histopaque-1083, 3-(4,5-dimethylthiazol-2-yl)-2,5-diphenyltetrazolium bromide (MTT) and dimethyl sulfoxide (DMSO) were purchased from Sigma-Aldrich (St. Louis, MO, USA). All reagents used in this study were of analytical grade.

### 4.2. Animals

Four-week-old female BALB/c nude mice and four-week-old male ICR mice were obtained from Orient Bio and NARA Biotech (Seoul, Korea), respectively. The mice were provided free access to commercial rodent chow and tap water and housed under specific pathogen-free conditions with a relative humidity of 50 ± 5% and a 12 h light/dark cycle at 22 ± 2 °C. All animal studies were conducted in accordance with experimental protocols approved by the Institutional Animal Care and Use Committee of the Yonsei University College of Dentistry (6 Jan 2015) and registered on the the Institutional Animal Care and Use Committee Yonsei University Health System (IACUC approval No. 2014-0016). All methods were carried out in accordance with the relevant guidelines and regulations.

### 4.3. Cell Culture

The human breast cancer cell lines MDA-MB-231 and MCF-7 were obtained from the Korean Cell Line Bank (Seoul, Korea). MDA-MB-231 and MCF-7 cells were cultured in DMEM supplemented with 10% FBS and 1% antibiotic-antimycotic mixture at 37 °C in a humidified atmosphere with 5% CO_2_. Human fetal osteoblastic hFOB1.19 cells were purchased from American Type Culture Collection (Manassas, VA, USA) and grown in DMEM/F-12 without phenol red containing 10% FBS, 0.3 mg/mL G418, and 1% antibiotic-antimycotic mixture at 34 °C in a humidified atmosphere with 5% CO2. Mouse bone marrow macrophages (BMMs) were isolated from the tibiae of four-week-old ICR male mice using Histopaque density gradient centrifugation and cultured in α-MEM containing 10% FBS, M-CSF (30 ng/mL), and 1% antibiotic-antimycotic mixture at 37 °C in a humidified atmosphere with 5% CO_2_, as previously described [41].

### 4.4. Cell Viability

MDA-MB-231 or MCF-7 breast cancer cells (2.5 × 10^3^ cells/well) were cultured in DMEM with chemerin at the indicated concentration in the absence or presence of TGF-β (10 ng/mL) or IGF-1 (40 ng/mL) for 24 h and 72 h. hFOB1.19 cells (5 × 10^3^ cells/well) were treated with DMEM/F-12 containing various concentrations of chemerin for 6 h and 24 h. BMMs (5 × 10^4^ cells/well) were treated with α-MEM containing chemerin at the indicated concentration in serum-free media for 5 days. Cell viability was measured with an MTT assay.

### 4.5. Cell Migration and Invasion

Cell migration and invasion assays were conducted using a Transwell chamber (Corning Costar, Cambridge, MA, USA) containing a polycarbonate membrane filter (6.5 mm diameter, 8 µm pore size), as previously described [41]. The lower surface of the insert membrane was precoated with gelatin (1 mg/mL) for the migration and invasion assays, and the upper surface was precoated with Matrigel (1 mg/mL; BD Biosciences, San Jose, CA, USA) for the invasion assay. MDA-MB-231 or MCF-7 cells (5 × 10^4^ cells/100 µL) were added into the insert of each Transwell chamber, which contained chemerin at the indicated concentration in the absence or presence of TGF-β (10 ng/mL) or IGF-1 (40 ng/mL). The lower chamber contained 600 µL of culture media containing 5% FBS and chemerin. After 8 h of incubation for the migration assay or 24 h of incubation for the invasion assay, the migrated or invaded cells were counted using a Zeiss Axio Imager microscope (Carl Zeiss, Gottingen, Germany).

### 4.6. CAM Assay

MDA-MB-231 or MCF-7 cells were labeled with CFDA SE Cell Tracer Kit (Invitrogen, Carlsbad, CA, USA) according to the manufacturer’s instructions. Fertilized chicken eggs (Pulmuone Co, Seoul, Korea) were incubated in a humidified incubator at 37 °C. Two days later, 3 mL of egg albumin was removed, and a window was made in the egg under aseptic conditions. The window was resealed with adhesive tape, and the eggs were further incubated to induce chick embryo development. On day 10, CFDA-SE-labeled MDA-MB-231 or MCF-7 cells were resuspended in a 4:1 mixture of Opti-MEM:Matrigel with chemerin at the indicated concentration. The suspended cells were placed onto the CAMs of the fertilized eggs (*n* = 4), and the resealed eggs were incubated for 3 days. On day 13, the CAMs were harvested, fixed with 4% paraformaldehyde for 24 h, and embedded in paraffin. The images were collected using a Zeiss LSM 700 confocal microscope (Zeiss Laboratories, Jena, Germany) and analyzed using ImageJ software. Cell invasion was determined by measuring the mean fluorescence of cells that had invaded below the CAM surface.

### 4.7. Preparation of Conditioned Medium of MDA-MB-231 Cells

MDA-MB-231 cells (1 × 10^6^ cells/100 mm dish) were cultured in DMEM containing 10% FBS for 24 h and then in serum-free DMEM/F12 for 24 h. The culture medium was centrifuged, and the supernatant was collected as the conditioned medium of MDA-MB-231 cells.

### 4.8. Western Blot Analysis

MDA-MB-231 or MCF-7 cells (1 × 10^6^ cells/100 mm dish) were treated with chemerin at the indicated concentration for 24 h. hFOB1.19 cells (1 × 10^6^ cells/100 mm dish) were treated with 75% conditioned medium from MDA-MB-231 cells and chemerin at the indicated concentrations for 6 h. Total cell lysates were prepared using RIPA buffer containing protease inhibitor cocktail (Santa Cruz Biotechnology) and centrifuged at 22,000 g for 15 min at 4 °C. The nuclear and cytosolic fractions were prepared using a nuclear/cytosol fractionation kit (BioVision, Mountain View, CA, USA) according to the manufacturer’s protocol. The protein concentration was quantitated using BCA protein assay reagents (Pierce, Rockford, IL, USA). Protein (40 µg) was subjected to 10% sodium dodecyl sulfate (SDS)-polyacrylamide gel electrophoresis and electrotransferred to a polyvinylidenedifluoride membrane (Millipore, Danvers, MA, USA). The membrane was blocked with 5% skim milk in TBS-T (10 nM Tris, pH 8.0, 150 mM NaCl, and 0.1% Tween-20) and then incubated with specific primary antibodies against the target proteins in 3% BSA in TBS-T. After washing with TBS-T, the membrane was incubated with horseradish peroxidase-conjugated secondary antibodies in 3% skim milk in TBS-T for 1 h at room temperature. The targeted proteins were visualized with Amersham ECL Western Blotting Detection Reagents (GE Healthcare, Little Chalfont, UK).

### 4.9. Gelatin Zymography

MDA-MB-231 or MCF-7 cells (1 × 10^6^ cells/100 mm dish) were incubated in 10% FBS-DMEM for 24 h. The cells were cultured in serum-free media containing chemerin at the indicated concentrations for an additional 24 h. After centrifugation, the supernatants were collected as the conditioned media. The protein concentration in the conditioned media was determined using a BCA Protein Assay Kit. The protein (20 µg) was subjected to electrophoresis in a 8% SDS-polyacrylamide gel containing 0.2% (*w*/*v*) gelatin. After electrophoresis, the gel was washed twice with 2.5% Triton X-100 for 1 h at room temperature and incubated in buffer containing 50 mM Tris-HCl (pH 7.4), 0.02% NaN_3_, 10 mM CaCl_2_, and 150 mM NaCl for 24 h at 37 °C. The gel was stained with a solution of 0.1% Coomassie brilliant blue R-250 (Flukachemie, AG, NeuUlm, Switzerland). The presence of clear zones within the blue background indicated the gelatinolytic activities of the MMPs.

### 4.10. Immunofluorescence Staining and Confocal Imaging

MDA-MB-231 or MCF-7 cells (2.5 × 10^3^ cells/well) were incubated in a chamber slide with complete medium for 24 h. The cells were treated with chemerin (80 nM) or SB525334 (TGF-β type I receptor inhibitor, 20 μM: Selleckchem, Houston, USA) in the presence of TGF-β (10 ng/mL) for 24 h. In addition, the cells were treated with chemerin (80 nM), ZSTK474 (PI3K inhibitor, 20 μM: Selleckchem), or PD98059 (MEK1/2 inhibitor, 20 μM: InvivoGen, San Diego, USA) in the presence of IGF-1 (40 ng/mL) for 24 h. The cells were fixed with 4% paraformaldehyde and permeabilized with Triton X-100 buffer. After blocking with 2% goat serum in PBS, the cells were incubated with specific primary antibodies against β-catenin, E-cadherin, or SMAD2/3 overnight at 4 °C. After washing, the cells were incubated with Alexa Fluor 488 goat anti-mouse IgG and Alexa Fluor 594 goat anti-rabbit IgG (Invitrogen) for 1 h at room temperature. The slide was mounted using Vectashield mounting medium with DAPI (Vector Laboratories, CA, USA). The images were collected using a Zeiss LSM 700 confocal microscope.

### 4.11. Enzyme-Linked Immunosorbent Assay (ELISA)

hFOB1.19 cells (5 × 10^4^ cells/well) were incubated in serum-free media containing chemerin at the indicated concentrations in the presence of 75% conditioned medium from MDA-MB-231 cells for 24 h. RANKL and OPG levels in the culture media were determined using commercially available ELISA kits (EIAab, Wuhan, China) according to the manufacturer’s instructions.

### 4.12. Osteoclast Formation Assay

BMMs were seeded at a density of 5 × 10^4^ cells/well in 96-well plates with α-MEM containing 10% FBS and treated with chemerin at the indicated concentrations in the presence of M-CSF (30 ng/mL) and RANKL (100 ng/mL) for 5 days. The culture medium was replaced with fresh media every second day. The cells were fixed with 3.7% formaldehyde for 1 min, and tartrate-resistant acid phosphatase (TRAP) was stained using the Acid Phosphatase Leukocyte kit (Sigma-Aldrich). The number of osteoclasts was determined by counting the multinucleated (more than 3 nuclei) TRAP-positive cells under a light microscope (original magnification, 40×).

### 4.13. Bone Resorption Activity

BMMs (5 × 10^4^ cells/well) in an Osteo Assay Stripwell Plate (Corning Costar, Cambridge, MA, USA) were cultured in α-MEM containing 10% FBS in the presence of M-CSF (30 ng/mL) and RANKL (100 ng/mL) for 5 days to induce their differentiation into mature osteoclasts. The BMMs were then treated with chemerin at the indicated concentrations in the presence of M-CSF (30 ng/mL) and RANKL (100 ng/mL) for an additional 2 days. Media were collected for gelatin zymography and a cathepsin K assay. BMMs were lysed with 5% sodium hypochlorite solution. The images of resorbed pits were obtained by using light microscopy (original magnification, 100×), and the area of the resorbed pits was measured with IMT i-Solution software (version 7.3, IMT i-Solution, BC, Canada). Cathepsin K levels in the media were measured using a SensiZyme cathepsin K activity assay kit (Sigma Aldrich) according to the manufacturer’s instructions. MMP levels were detected by gelatin zymography.

### 4.14. Tumor Xenograft Model

Five-week-old female BALB/c nude mice were randomly divided into 4 groups, with 5 mice per group, consisting of the control and MCF-7 cell-inoculated groups treated with chemerin at the indicated doses. MCF-7 cells were resuspended in an HBSS:Matrigel (1:1) mixture, and chemerin was dissolved in 0.1% bovine serum albumin (BSA) in PBS. Mice were anesthetized via i.p. injection of a Zoletil (30 mg/kg)/Rompun (10 mg/kg) mixture. One day prior to cancer cell injection, mice were subcutaneously implanted with 17β-estradiol pellets (0.72 mg, 60-day release; Innovative Research of America, Sarasota, FL, USA). The resuspended MCF-7 cells (1 × 10^7^ cells/0.1 mL) were subcutaneously injected into the right flanks of the mice. Control mice received the HBSS:Matrigel (1:1) mixture (0.1 mL) alone. After twenty-four hours, chemerin at the indicated doses was intraperitoneally injected every two days for 6 weeks. The tumor volume was measured with a digital electric caliper and calculated in accordance with the following formula: (length × width^2^)/2.

### 4.15. Murine Model of Breast Cancer-Induced Osteolysis

Five-week-old female BALB/c nude mice were randomly divided into 5 groups, with 7 mice per group. Mice were anesthetized with a Zoletil (30 mg/kg)/Rompun (10 mg/kg) mixture. MDA-MB-231 cells (1 × 10^6^ cells/50 µL HBSS) were injected into the medullary cavity of the proximal tibia of the mice. Control mice received HBSS alone. After twenty-four hours, chemerin at the indicated doses in 0.1% BSA was intraperitoneally injected every two days for 5 weeks. On day 35, the collected tibiae were scanned with a μCT system (SkyScan 1076, SkyScan, Aartselaar, Belgium). Three-dimensional (3D) images were generated using NRecon software (SkyScan). Serum levels of calcium, CTX, and TRAP 5b were determined using a QuantiChrome calcium assay kit (BioAssay Systems, Hayward, CA, USA), a mouse TRAP assay kit (Immuno Diagnostic Systems, Boldon, UK), and a RatLaps enzyme immunoassay kit (Immuno Diagnostic Systems), respectively.

### 4.16. Statistics

Data are expressed as the mean ± standard error (SE) of three independent experiments. Statistical analysis was performed with one-way ANOVA and Student’s *t*-test to determine the difference between the two groups. All statistical analyses were performed using SPSS version 19.0 (SPSS Inc., New York, NY, USA). *p* < 0.05 was considered statistically significant.

## Figures and Tables

**Figure 1 ijms-21-02871-f001:**
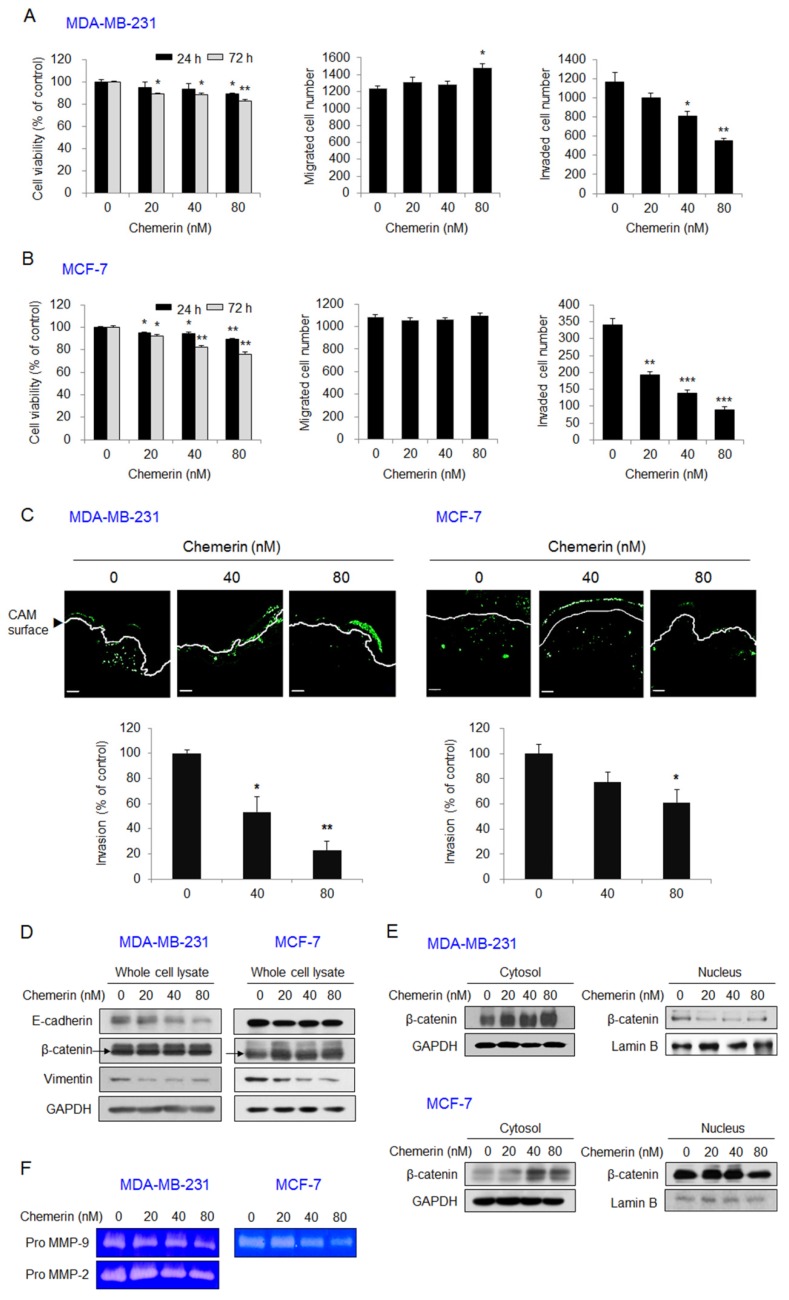
Chemerin suppressed the viability, invasion, and epithelial–mesenchymal transition (EMT) of breast cancer cells. (**A**,**B**) The viability, migration, and invasion of (**A**) MDA-MB-231 or (**B**) MCF-7 breast cancer cells treated with chemerin. Cell viability was measured by an 3-(4,5-dimethylthiazol-2-yl)-2,5-diphenyltetrazolium bromide (MTT) assay after 24 h and 72 h of treatment with chemerin. Cell migration or invasion was measured using a Transwell chamber and determined by counting the number of migrated or invaded breast cancer cells 8 h or 24 h after chemerin treatment. (**C**) In ovo invasion of chemerin-treated MDA-MB-231 or MCF-7 cells. CFDA-SE-labeled MDA-MB-231 or MCF-7 cells were loaded onto the chorioallantoic membrane (CAM) of fertilized eggs (*n* = 4) and incubated for 72 h. The images were collected using a Zeiss LSM 700 confocal microscope and analyzed using ImageJ software. Representative images (upper). Scale bar, 100 μm. Cell invasion was determined by measuring the mean fluorescence of cells that had invaded below the CAM surface (lower); (**D**) The expression levels of EMT markers and (**E**) the nuclear and cytosolic levels of β-catenin in MDA-MB-231 or MCF-7 cells treated with chemerin for 24 h. The expression level of E-cadherin, β-catenin, or vimentin in the whole cell lysate and the nuclear and cytosolic levels of β-catenin were detected using Western blotting. Representative images; (**F**) The levels of pro matrix metalloproteinase (MMP)-2 and pro MMP-9 secreted from MDA-MB-231 or MCF-7 cells treated with chemerin for 24 h. The levels of pro MMPs in the collected conditioned media were determined by gelatin zymography. The clear zones in representative images indicate the gelatinolytic activity of the MMPs. Data are expressed as the mean ± SEM. * *p* < 0.05, ** *p* < 0.01, *** *p* < 0.001 versus cells without chemerin.

**Figure 2 ijms-21-02871-f002:**
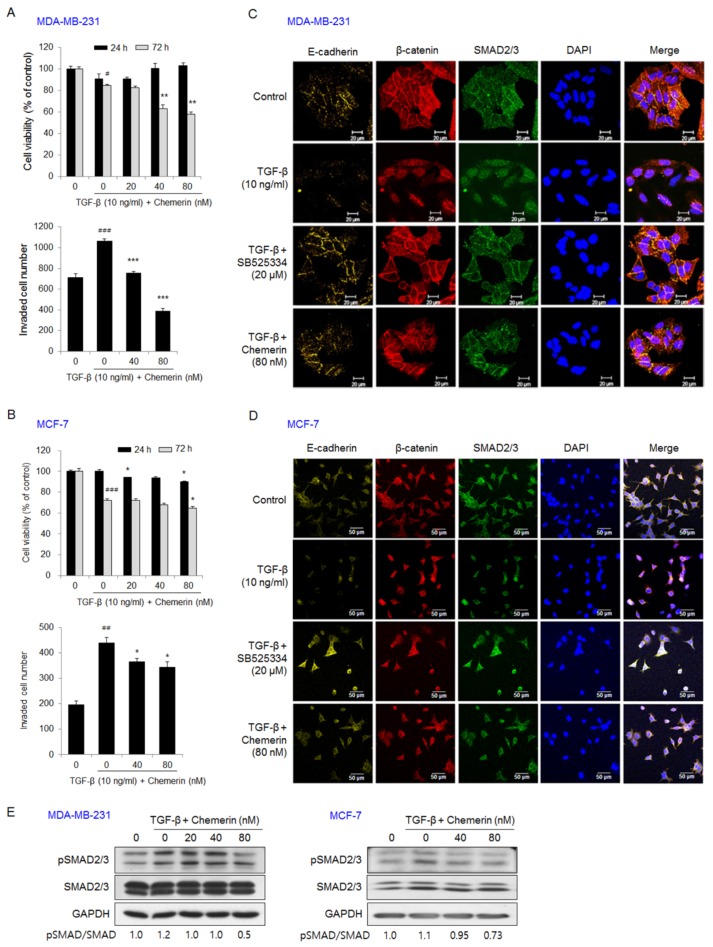
Chemerin suppressed the viability, invasion, and EMT of transforming growth factor (TGF)-β-treated breast cancer cells. (**A**,**B**) The viability and invasion of (A) MDA-MB-231 or (B) MCF-7 cells treated with chemerin in the presence of TGF-β. Cell viability was measured by an MTT assay after 24 h and 72 h of treatment with TGF-β and chemerin. Cell invasion was measured using a Transwell chamber and determined by counting the number of invaded MDA-MB-231 and MCF-7 cells 24 h after TGF-β and chemerin treatment. (**C**,**D**) The expression levels and cellular localization of E-cadherin, β-catenin, and smad2/3 in (**C**) MDA-MB-231 and (**D**) MCF-7 cells treated with chemerin or SB525334 (TGF-β type I receptor inhibitor) in the presence of TGF-β for 24 h. The cells were immunostained with specific primary antibodies against β-catenin, E-cadherin, or SMAD2/3 and then with Alexa Fluor 488 goat anti-mouse IgG or Alexa Fluor 594 goat anti-rabbit IgG. The slides were mounted with Vectashield mounting medium with 4′,6-diamidino-2-phenylindole (DAPI). The images were collected using a Zeiss LSM 700 confocal microscope. Representative immunofluorescence images. Scale bar, 20 μm in MDA-MB-231 cells and 50 μm in MCF-7 cells; (**E**) The expression levels of SMAD2/3 and pSMAD2/3 in MDA-MB-231 or MCF-7 cells treated with chemerin in the presence of TGF-β for 24 h. The expression levels were detected using Western blotting. Representative images. Data are expressed as the mean ± SEM. # *p* < 0.05, ## *p* < 0.01, ### *p* < 0.01 versus cells without TGF-β and chemerin, * *p* < 0.05, ** *p* < 0.01, *** *p* < 0.005 versus cells treated with TGF-β alone.

**Figure 3 ijms-21-02871-f003:**
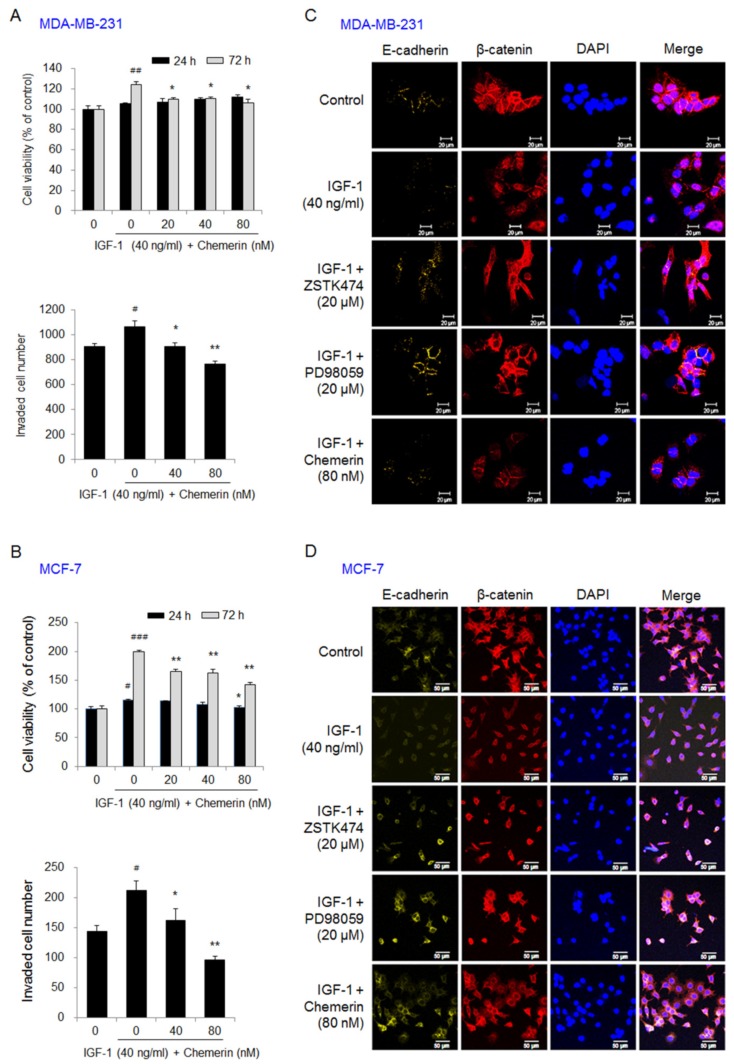
Chemerin suppressed the viability, invasion, and EMT of insulin-like growth factor (IGF)-1-treated breast cancer cells. (**A**,**B**) The viability and invasion of (**A**) MDA-MB-231 or (**B**) MCF-7 cells treated with IGF-1 and chemerin for 24 and 72 h. Cell viability was measured by an MTT assay. Cell invasion was determined by counting the number of invaded MDA-MB-231 and MCF-7 cells 24 h after IGF-1 and chemerin treatment using a Transwell chamber; (**C**,**D**) The expression levels and cellular localization of E-cadherin and β-catenin in MDA-MB-231 or MCF-7 cells treated with chemerin, ZSTK474 (PI3K inhibitor), or PD98059 (MEK1/2 inhibitor) in the presence of IGF-1. The cells were immunostained with specific primary antibodies against β-catenin or E-cadherin and then with Alexa Fluor 488 goat anti-mouse IgG or Alexa Fluor 594 goat anti-rabbit IgG. The slides were mounted using Vectashield mounting medium with DAPI. The images were collected using a Zeiss LSM 700 confocal microscope. Representative immunofluorescence images. Scale bar, 20 μm in MDA-MB-231 cells and 50 μm in MCF-7 cells. Data are expressed as the mean ± SEM. # *p* < 0.05, ## *p* < 0.01, ### *p* < 0.001 versus cells without IGF-1 and chemerin, * *p* < 0.05, ** *p* < 0.01 versus cells treated with IGF-1 alone.

**Figure 4 ijms-21-02871-f004:**
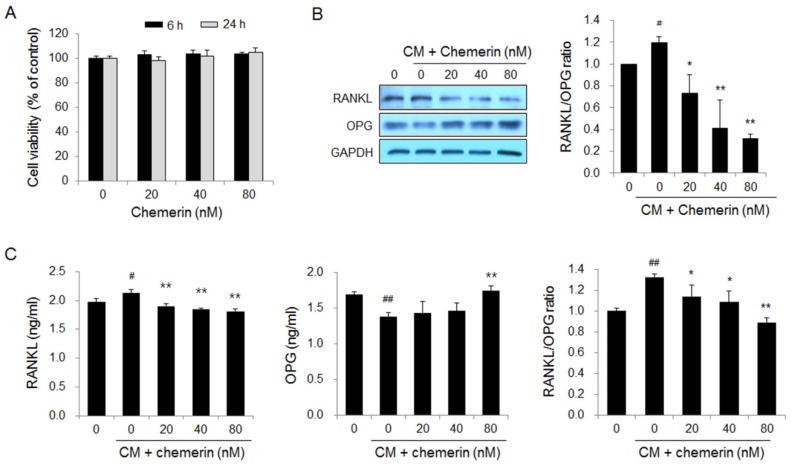
Chemerin suppressed breast cancer cell-induced increase in the RANKL/OPG ratio in osteoblastic cells. (**A**) The viability of hFOB1.19 osteoblastic cells treated with chemerin for 6 and 24 h; (**B**) Protein expression and (**C**) secretion levels of RANKL and OPG in hFOB1.10 osteoblastic cells treated with chemerin in the presence of the conditioned medium (CM) of MDA-MB-231 cells for 6 h and 24 h, respectively. The expression and secretion levels of RANKL and OPG proteins were detected by Western blotting and ELISA, respectively. # *p* < 0.05, ## *p* < 0.01 versus cells without CM and chemerin, * *p* < 0.05, ** *p* < 0.01 versus cells treated with CM alone.

**Figure 5 ijms-21-02871-f005:**
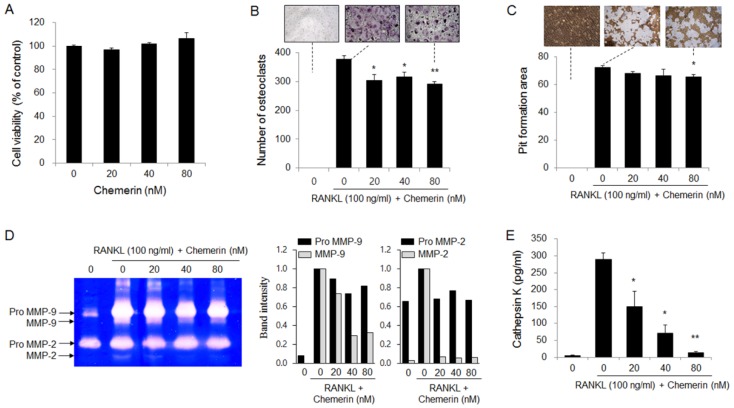
Chemerin slightly suppressed RANKL-induced osteoclast formation and activity in bone marrow-derived macrophages (BMMs). (**A**) The viability of BMMs treated with chemerin for 5 days; (**B**) The number of differentiated osteoclasts derived from BMMs treated with chemerin in the presence of macrophage colony-stimulating factor (M-CSF) and RANKL for 5 days. The cells were stained for tartrate-resistant acid phosphatase (TRAP), and the multinucleated (more than 3 nuclei) TRAP-positive cells were counted as osteoclasts under a light microscope. Representative images at 40× magnification; (**C**) The formation of resorption pits by osteoclasts. Representative images at 100× magnification; (**D**) MMP levels secreted from osteoclasts; (**E**) Cathepsin K levels secreted from osteoclasts. (**C**–**E**) BMMs were treated with M-CSF and RANKL in Osteo Assay Stripwell Plates for 5 days and then with chemerin in the presence of M-CSF and RANKL for an additional 2 days. BMMs were lysed, and the area of resorbed pits was calculated with IMT i-Solution software. The levels of pro and active MMP-2 and MMP-9 in the culture media were detected with gelatin zymography. Band intensities were measured with Image J software. Cathepsin K levels in the culture media were measured using an ELISA kit. Data are expressed as the mean ± SEM. * *p* < 0.05, ** *p* < 0.01 versus cells treated with RANKL alone.

**Figure 6 ijms-21-02871-f006:**
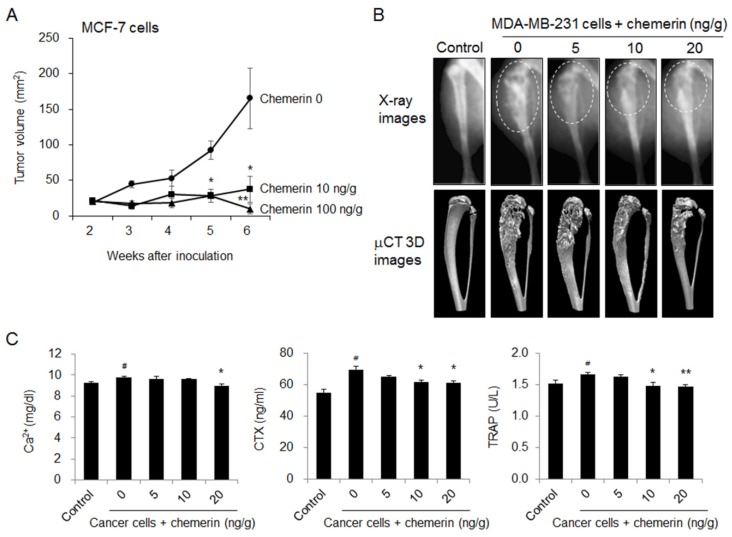
Chemerin suppressed tumor growth and breast cancer cell-associated bone loss in mice. (**A**) Tumor growth in mice inoculated with MCF-7 breast cancer cells in the presence of a β-estradiol pellet and then intraperitoneally administered with chemerin for 6 weeks. * *p* < 0.05, ** *p* < 0.01 versus MCF-7 cell-inoculated mice; (**B**) Representative radiophotographic and μCT 3D images of osteolytic lesions (dotted circles) and (**C**) serum levels of bone turnover markers in mice with intratibial inoculation of MDA-MB-231 breast cancer cells followed by intraperitoneal administration of chemerin for 5 weeks. # *p* < 0.05 versus control mice, * *p* < 0.05, ** *p* < 0.01 versus MDA-MB-231 cell-inoculated mice.

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
