# Peer review of "Chemerin Treatment Inhibits the Growth and Bone Invasion of Breast Cancer Cells"

_ijms, 2020, doi:10.3390/ijms21082871_

Round 1

Author Response

*Abstract: It should be more concise. “Chemerin treatment inhibited nuclear β-catenin levels in breast cancer cells stimulated with or not treated with TGF-β or IGF-1”. “Stimulated with or not treated with” sounds confusing. Could the authors please re-phrase this?

--> We repharased “Stimulated with or not treated with” to ‘stimulated with or without’.

*Introduction:

- Line 33-CMKLR1: could the authors please spell this out?

--> In Abbreviations, we described the full name of CMKLR1 as chemerin/chemokine-like receptor 1.

- In the manuscript, the authors focus on the effect of chemerin on bone in the context of breast cancer, however in the introduction, breast cancer bone metastasis, bone turnover and vicious cycle are not mentioned. Could the authors describe this in more details?

--> In this study, we investigated the inhibitory effect of chemerin on the viability, invasion, EMT of weakly metastatic MCF-7 and strongly metastatic, bone-tropic MDA-MB-231 cells. To further determine whether chemerin treatment could also inhibit breast cancer-mediated bone loss, we investigated the effects of chemerin on the production of osteoblastic RANKL and OPG, RANKL-induced osteoclastogenesis, and bone resorption. Thus, we described the vicious cycle of breast cancer bone metastasis in the Discussion section, not in the Introduction section, as follows. ‘In breast cancer bone metastasis, the release of osteolytic factors from breast cancer cells exacerbates cancer cell-mediated bone loss by increasing the RANKL/OPG ratio in osteoblastic cells and consequently causing osteoclast-mediated bone resorption [37]’.

*Results:

- Figure 1A-migrated cell number: is the increase statistically significant? If not, then statements in the results section (line 82, 104, 105) might need reviewing.

--> The migrated cell number was statistically significantly increased in MDA-MB-231 cells treated with 80 nM chemerin. We denoted statistical significance in Figure 1A-Migrated cell number.

- Line 87-89: English to be checked

In the in ovo model, in the quantification of invading cells, have the authors taken into account the size of each object? Because tumour cells are labelled with a fluorescent dye, is it possible that upon tumour cell death, cell fragments are taken up by immune cells?

--> We labeled cancer cells with a fluorescent dye CFDA-SE and cultured the labeled cancer cells. Then, the labeled cancer cells were loaded on CAM. The invaded cells were quantified by measuring the labeled cancer cell-derived fluorescence. The size of detected fluorescent spot (fluorescence intensity) is proportional to the number of cells.

Under this experimental condition, the cells do not die.

- Line 98: As indicated by the authors, Vimentin is a mesenchymal marker. Therefore, Vimentin downregulation does not support a shift towards EMT. Could the author look at other mesenchymal markers in addition to Vimentin and B-catenin? N-cadherin is another mesenchymal marker.

--> Because vimentin is a mesenchymal marker, vimentin downregulation by chemerin treatment supports that chemerin treatment inhibited EMT. Although we expected an epithelial marker E-cadherin to increase because EMT was inhibited by chemerin treatment, E-cadherin was decreased by chemerin treatment in two breast cancer cell lines, particularly in MDM-MB-231 cells, in the absence of TGF-beta and IGF-1.

In our preliminary study, chemerin treatment did not change the expression levels of E-cadherin-regulating transcription factors, Snail, Twist, and Zeb1/2. The decrease in E-cadherin expression may be due to its degradation by any proteases but we could not find. We suggested that the decreased in E-cadherin expression is associated with the increased migration of MDA-MB-231 cells.

   The inhibitory effect of chemerin on EMT was supported by the downregulated vimentin expression, the decreased nuclear b-catenin level, and the reduced MMP secretion levels in chemerin-treated breast cancer cell lines. The secretion of MMPs is also increased in EMT.

- Line 100-102: Is it possible that chemerin affects the expression level on MMP9 and MMP2 in addition to the secretion of pro-MMP2 and pro-MMP9? Could the authors perform a qpcr to verify that? Also because the decrease in pro-MMP9 and pro-MMP2 is minimal.

Figure 1F: The authors mention that it represents the levels of MMP secreted and the activities of MMP2 and MMP9? Could the authors specify which one of the two is it, level or secretion?

--> We changed black and white Figure 1F to color. Chemerin treatment inhibited the levels of pro MMP-2 and pro MMP-9 in the conditioned media of breast cancer cells.

In Figure 1F, we corrected ‘The activities of MMP-2 and MMP-9’ to ‘The levels of pro MMPs’.

- Figure 1D compared to Figure 2C (tgfb+chemerin): Chemerin treatment inhibits E-cadherin expression (Figure 1D), whilst increases E-cadherin expression and rescues E-cadherin inhibition after TGF-B treatment (Figure 2C). Could the authors comment on how the two data are in agreement?

--> As mentioned above, although we expected an epithelial marker E-cadherin to increase because EMT was inhibited by chemerin treatment, E-cadherin was decreased by chemerin treatment in two breast cancer cell lines, particularly in MDM-MB-231 cells, in the absence of TGF-beta and IGF-1 (Figure 1D).

However, chemerin treatment rescued the decreased E-cadherin expression in TGF-b-treated breast cancer cells (Figure 2C). TGF-b significantly decreased E-cadherin expression and chemerin treatment inhibited a decrease in E-cadherin expression by blocking the translocation of SMAD2/3 in TGF-b-treated breast cancer cells. Chemerin effect blocking TGF-b-induced reduction of E-cadherin expression may be more potent than the effect of chemerin itself reducing E-cadherin expression level.

- Figure 2A (invasion): Chemerin inhibits TGF-B induced invasion and displays an inhibitory effect on cell invasion by itself (80 nM); Figure 2B (Invasion): Chemerin inhibits TGF-B induced invasion, but does not display an inhibitory effect on cell invasion by itself (80 nM). Could the authors comment on why there is a difference between MDA-MB-231 and MCF-7?

--> In Figure 2A and 2B, chemerin inhibited TGF-b-induced invasion in MDA-MB-231 and MCF-7 cells. Chemerin itself inhibited cell invasion in both breast cancer cell lines as shown in Figure 1A and 1B. Treatment with 80 nM chemerin seems to inhibit better TGF-b-induced invasion in MDA-MB-231 cells than in MCF-7 cells. This result may be related with the inhibitory effect of chemerin at 80 nM on TGF-b-induced phosphylation of SMAD2/3 in MDA-MB-231 cells than in MCF-7 cells.

- Lines 193-195: I agree with the authors that the RANKL/OPG ratio increases upon CM treatment and decreases when chemerin is added. I would not say that the WB shows an increase in RANKL and decrease in OPG upon CM treatment as the bands look very similar to the untreated group.

--> We changed Western blotting images. To support Figure 1B, we measured the secreted RANKL and OPG levels into the conditioned media of CM-treated osteoblastic cells using ELISA kits (Figure 1C).  

Have the authors checked if chemerin has an effect on RANKL and OPG also in MCF-7 cells? MCF-7 is generally used as a model for studying ER+ breast cancer, which generally forms metastasis in bone more frequently than triple negative breast cancer (modelled by using MDA-MB-231)

--> Triple negative breast cancer MDA-MB-231 cells are highly metastatic and primarily metastasize into bone. ER+ MCF-7 cells are less metastatic. In our preliminary experiment, MCF-7-derived conditioned medium did not affect the expression and production of RANKL and OPG, while MDA-MB-231-derived conditioned medium increased RANKL expression and production and reduced OPG expression and production.

- Figure 5B and C: in the untreated control, are there no osteoclasts at all and no pit formation at all? Based on Figure 5F, Cathepsin K can be detected from osteoclasts, so it should be possible to detect some osteoclasts.

--> In the untreated control, there are no osteoclasts at all and no pit formation. In the presence of RANKL, BMMs as osteoclast precursors differentiate into mature osteoclasts.

We induced osteoclast differentiation by RANKL treatment for 5 days and further incubated osteoclasts in the presence of RANKL and chemerin. Then, we detected cathepsin K level in the culture media using ELISA.

- Figure 5D: How was the secretion of MMP9 measured? Could the authors provide a quantification of the bands? I agree with a reduced secretion of MMP2 upon chemerin treatment, but I cannot see a difference for MMP9.

--> We induced osteoclast differentiation by RANKL treatment for 5 days and further incubated osteoclasts in the presence of RANKL and chemerin for 2 days. Then, we detected the MMP levels in the culture media by gelatin zymography. MMP-2 and MMP-9 levels were weakly decreased by chemerin treatment. We changed black and white Figure 5D to color.

- In vivo experiment: Can the authors comment on why MCF-7 cells have been inoculated subcutaneously instead of intra-tibial? Did the authors look at P1NP as a bone turnover marker? Looking at P1NP would inform on osteoblast activity in vivo.

--> MCF-7 cells are less metastatic and b-estradiol release is essential for the growth of MCF-7 cells. We used MCF-7 cells in tumor xenograft model to estimate chemerin effect on breast cancer cell growth and MDA-MB-231 cells in intratibial model to estimate chemerin effect on breast cancer cell-mediated bone damage.

We did not measure PINP.

In the serum of mice inoculated with bone invasive cancer cells, the levels of bone formation markers, as well as bone resorption markers, are also increased. The increased levels of bone formation markers may cause as a result corresponding to the increased bone resorption. However, we did not measure bone formation markers in this study.

*Discussion:

- In the discussion, the first paragraph refers to adipocytes. The authors mentioned in the introduction that chemerin has been reported to be an adipokine, however there is no work in their manuscript that relates to it. Therefore, I would not start the discussion with information regarding the adipose tissue.

Could the authors put the greek letter β in TGF- β and β-Catenin?

--> We deleted the introduction that chemerin has been reported to be an adipokine. We put the greek letter β in TGF- β and β-Catenin.

- In the introduction, the authors mention that chemerin is produced by different cell types. It is interesting to see from the authors’ work that chemerin has an effect on both cancer cells and bone cells. Is the source of chemerin known in bone?

--> Bone marrow adipocytes are one of the most abundant cell types found in bone marrow tissue. They constitute approximately 15% of the bone marrow volume in young adults, rising to 60% by the age of 65 years old. Chemerin can release adipocytes in bone-tumor microenvironment.

- The authors mention in the discussion that chemerin is produced as prochemerin and then cleaved to be activated. Could the authors provide information about production and activation of chemerin in bone, if references are available?

--> We provide two references. Chemerin in bone may be released from bone marrow adipocytes and mesenchymal stromal cells.

  1. Buechler C, Feder S, Haberl EM, Aslanidis C. Chemerin Isoforms and Activity in Obesity. Int J Mol Sci. 2019 Mar 5;20(5).
  2. Vinci P, Bastone A, Schiarea S, Cappuzzello C, Del Prete A, Dander E, Biondi A, D'Amico G. Mesenchymal stromal cell-secreted chemerin is a novel immunomodulatory molecule driving the migration of ChemR23-expressing cells. Cytotherapy. 2017 Feb;19(2):200-210.

- In the discussion, the sentence “chemerin prevents breast cancer cell-associated bone loss by suppressing breast cancer cell invasion via EMT” is a bit an overstatement. The data show an effect of chemerin on EMT and on bone loss, indeed, however there is no evidence that there is a cause/effect between the two. I would modify this sentence by saying that chemerin prevents EMT and bone loss. (Same comment for the last sentence in the abstract).

--> We corrected in the discussion as follows: chemerin inhibits the growth and invasion of breast cancer cells. In addition, chemerin prevents breast cancer cell-mediated bone loss by reducing the RANKL/OPG ratio in osteoblastic cells, and inhibiting osteoclast formation and activity.

We corrected in the Abstract as follows: Taken together, chemerin inhibits the growth and invasion of breast cancer cells and prevents bone loss resulting from breast cancer cells by inhibiting finally osteoclast formation and activity.

Reviewer 2 Report

This is an interesting study. Major points that the authors need to address are as follows:

  1. The molecular mechanism(s) by which chemerin can affect the viability and invasion of breast cancer cells should be investigated in detail? For example, whether deletion of beta-catenin by si-RNA can affect the observed anti-cancer effects of chemerin.
  2. Acute toxicity studies should be performed to establish the safety of chemerin.
  3. The authors should provide their own justification and relevance of the study. This will help the readers to understand the importance of the paper.
  4. As transcription factor NF-κB plays an important role in regulating MMPs expression, it will be interesting if the authors can observe the effect of chemerin on NF-κB signaling. Also, relevant literature such as (PMID: 25083991; PMID: PMID: 18566231 can be included in the discussion section of the article.
  5. Typographical errors were found throughout the manuscript and should be corrected.

Author Response

Responses to Reviewers’ comments

We revised our manuscript according to Reviewers’ comments and highlighted the corrected parts using the "Track Changes" function in Microsoft Word.

Reviewer 2

  1. The molecular mechanism(s) by which chemerin can affect the viability and invasion of breast cancer cells should be investigated in detail? For example, whether deletion of beta-catenin by si-RNA can affect the observed anti-cancer effects of chemerin.

--> In this study, we focused on chemerin effect on breast cancer cell invasion and related bone loss. We suggest that anti-invasive activity of chemerin is associated with its EMT inhibition effect and its anti-bone resorptive effect is associated with the reduced osteoblastic RANKL/OPG ratio and osteoclast activity. In further study, we will determine the molecular mechanisms underlying chemerin effects.

  1. Acute toxicity studies should be performed to establish the safety of chemerin.

--> We will investigate acute toxicity as one of next further studies. In this study, we focused on chemerin effect on breast cancer cell invasion and related bone loss.

  1. The authors should provide their own justification and relevance of the study. This will help the readers to understand the importance of the paper.

--> The role of chemerin in breast cancer is controversial. In the present study, we investigated the effect of chemerin on breast cancer cell growth and invasion and related bone resorption.

   In line 66-76, we attempted to provide our justification and relevance of this study.

  1. As transcription factor NF-κB plays an important role in regulating MMPs expression, it will be interesting if the authors can observe the effect of chemerin on NF-κB signaling. Also, relevant literature such as (PMID: 25083991; PMID: PMID: 18566231 can be included in the discussion section of the article.

--> We included discussion and references in line 284-286 as follows. NF-kB-regulated MMP-2 and MMP-9 secretion is also closely associated with EMT [37,38]. The secreted levels of pro MMP-2 and pro MMP-9 were reduced by chemerin treatment.

  1. Typographical errors were found throughout the manuscript and should be corrected.

--> We checked again.

Round 2

Author Response

- Line 87-89: In the in ovo model, in the quantification of invading cells, have the authors taken into account the size of each object? Because tumour cells are labelled with a fluorescent dye, is it possible that upon tumour cell death, cell fragments are taken up by immune cells?

--> We labeled cancer cells with a fluorescent dye CFDA-SE and cultured the labeled cancer cells. Then, the labeled cancer cells were loaded on CAM. The invaded cells were quantified by measuring the labeled cancer cell-derived fluorescence. The size of detected fluorescent spot (fluorescence intensity) is proportional to the number of cells.

Under this experimental condition, the cells do not die.

A marker of cell death should be used to mention that cells do not die. However, CFDA-SE dye appears to bind covalently to intra-cellular molecules. Hence, differently from other dyes, this would suggest that, upon cell death, the signal from CFDA-SE disappears and only viable cells should be labelled.

--> Cimpean et al. mentioned an advantage of CAM assay as follows: chick embryo is naturally immunodeficient and can therefore accept transplantation from various tissues and species without specific or nonspecific immune responses (Cimpean AM, Ribatti D, Raica M. The chick embryo chorioallantoic membrane as a model to study tumor metastasis. Angiogenesis. 2008;11(4):311-9.)

Invitrogen guides CFDA-SE Cell Tracer kit as follows: CFDA SE shows little cytotoxicity, with minimal observed effect on the proliferative ability or biology of cells. The dye–protein adducts that form in labeled cells are retained by the cells throughout development, meiosis, and in vivo tracing. The label is inherited by daughter cells after cell division, or cell fusion, and is not transferred to adjacent cells in a population. Lymphocytes labeled with CFDA SE have been detected up to eight weeks after injection into mice in lymphocyte-migration studies, and viable hepatocytes that were similarly labeled were easily located by fluorescence microscopy even 20 days after intrahepatic transplantation.

Wang et al. also indicated that CFDA SE is non-toxic (Wang XQ, Duan XM, Liu LH, Fang YQ, Tan Y. Carboxyfluorescein diacetate succinimidyl ester fluorescent dye for cell labeling. Acta Biochim Biophys Sin (Shanghai). 2005, 37, 379-385).

- Line 98: As indicated by the authors, Vimentin is a mesenchymal marker. Therefore, Vimentin downregulation does not support a shift towards EMT. Could the author look at other mesenchymal markers in addition to Vimentin and B-catenin? N-cadherin is another mesenchymal marker.

--> Because vimentin is a mesenchymal marker, vimentin downregulation by chemerin treatment supports that chemerin treatment inhibited EMT. Although we expected an epithelial marker E-cadherin to increase because EMT was inhibited by chemerin treatment, E-cadherin was decreased by chemerin treatment in two breast cancer cell lines, particularly in MDM-MB-231 cells, in the absence of TGF-beta and IGF-1.

In our preliminary study, chemerin treatment did not change the expression levels of E-cadherin-regulating transcription factors, Snail, Twist, and Zeb1/2. The decrease in E-cadherin expression may be due to its degradation by any proteases but we could not find. We suggested that the decreased in E-cadherin expression is associated with the increased migration of MDA-MB-231 cells.

Could the authors mention this in the discussion?

--> We mentioned this in the Discussion (line 285-288).

The inhibitory effect of chemerin on EMT was supported by the downregulated vimentin expression, the decreased nuclear b-catenin level, and the reduced MMP secretion levels in chemerin-treated breast cancer cell lines. The secretion of MMPs is also increased in EMT.

Did the authors try to look at the ratio of mesenchymal and epithelial markers instead of considering them separately? It seems that most EMT markers are not affected by Chemerin.

--> In our study, it seems difficult to explain anti-invasive effect of chemerin with the ratio of mesenchymal and epithelial markers. The nuclear translocation of b-catenin and MMP secretion is associated with EMT. Chemerin treatment inhibited the nuclear translocation of b-catenin and MMP secretion.

In addition, could the author mention their sentence “The decrease in E-cadherin expression may be due to its degradation” in discussion?

 --> We mentioned this sentence in the Discussion (line 286-288).

- Figure 1D compared to Figure 2C (tgfb+chemerin): Chemerin treatment inhibits E-cadherin expression (Figure 1D), whilst increases E-cadherin expression and rescues E-cadherin inhibition after TGF-B treatment (Figure 2C). Could the authors comment on how the two data are in agreement?

--> As mentioned above, although we expected an epithelial marker E-cadherin to increase because EMT was inhibited by chemerin treatment, E-cadherin was decreased by chemerin treatment in two breast cancer cell lines, particularly in MDM-MB-231 cells, in the absence of TGF-beta and IGF-1 (Figure 1D).

However, chemerin treatment rescued the decreased E-cadherin expression in TGF-b-treated breast cancer cells (Figure 2C). TGF-b significantly decreased E-cadherin expression and chemerin treatment inhibited a decrease in E-cadherin expression by blocking the translocation of SMAD2/3 in TGF-b-treated breast cancer cells. Chemerin effect blocking TGF-b-induced reduction of E-cadherin expression may be more potent than the effect of chemerin itself reducing E-cadherin expression level.

Can the authors mention this in the discussion?

--> We mentioned in the Discussion (line 289-291).

- Figure 2A (invasion): Chemerin inhibits TGF-B induced invasion and displays an inhibitory effect on cell invasion by itself (80 nM); Figure 2B (Invasion): Chemerin inhibits TGF-B induced invasion, but does not display an inhibitory effect on cell invasion by itself (80 nM). Could the authors comment on why there is a difference between MDA-MB-231 and MCF-7?

--> In Figure 2A and 2B, chemerin inhibited TGF-b-induced invasion in MDA-MB-231 and MCF-7 cells. Chemerin itself inhibited cell invasion in both breast cancer cell lines as shown in Figure 1A and 1B. Treatment with 80 nM chemerin seems to inhibit better TGF-b-induced invasion in MDA-MB-231 cells than in MCF-7 cells. This result may be related with the inhibitory effect of chemerin at 80 nM on TGF-b-induced phosphylation of SMAD2/3 in MDA-MB-231 cells than in MCF-7 cells.

Can the authors add this in the discussion?

--> We included Western blot data as Figure 2E and mentioned in Result (line 141-142) and Discussion (line 275-278).

- Figure 5D: How was the secretion of MMP9 measured? Could the authors provide a quantification of the bands? I agree with a reduced secretion of MMP2 upon chemerin treatment, but I cannot see a difference for MMP9.

--> We induced osteoclast differentiation by RANKL treatment for 5 days and further incubated osteoclasts in the presence of RANKL and chemerin for 2 days. Then, we detected the MMP levels in the culture media by gelatin zymography. MMP-2 and MMP-9 levels were weakly decreased by chemerin treatment. We changed black and white Figure 5D to color.

I can appreciate that the coloured image helps, however, I would prefer to see a quantification of the bands

--> We measured band intensities with Image J software and added the plots in Figure 5D.

¢ Discussion:

In the introduction, the authors mention that chemerin is produced by different cell types. It is interesting to see from the authors’ work that chemerin has an effect on both cancer cells and bone cells. Is the source of chemerin known in bone?

--> Bone marrow adipocytes are one of the most abundant cell types found in bone marrow tissue. They constitute approximately 15% of the bone marrow volume in young adults, rising to 60% by the age of 65 years old. Chemerin can release adipocytes in bone-tumor microenvironment.

Could the authors add this information in the manuscript?

--> We added this information in the Discussion (line 299-302).

Round 3

Reviewer 1 Report

Thanks to the authors for their replies. My comments have been addressed.